# Chemical and Mechanical Aspect of Entropy-Exergy Relationship

**DOI:** 10.3390/e23080972

**Published:** 2021-07-28

**Authors:** Pierfrancesco Palazzo

**Affiliations:** Technip Energies, 00148 Roma, Italy; pierfrancesco.palazzo@technipenergies.com or pierfrancesco.palazzo@gmail.com

**Keywords:** Carnot cycle, Carnot efficiency, thermal entropy, chemical entropy, mechanical entropy, thermal exergy, chemical exergy, mechanical exergy, metabolic reactions

## Abstract

The present research focuses the chemical aspect of entropy and exergy properties. This research represents the complement of a previous treatise already published and constitutes a set of concepts and definitions relating to the entropy–exergy relationship overarching thermal, chemical and mechanical aspects. The extended perspective here proposed aims at embracing physical and chemical disciplines, describing macroscopic or microscopic systems characterized in the domain of industrial engineering and biotechnologies. The definition of chemical exergy, based on the Carnot chemical cycle, is complementary to the definition of thermal exergy expressed by means of the Carnot thermal cycle. These properties further prove that the mechanical exergy is an additional contribution to the generalized exergy to be accounted for in any equilibrium or non-equilibrium phenomena. The objective is to evaluate all interactions between the internal system and external environment, as well as performances in energy transduction processes.

## 1. Introduction

The research here present follows, and is complementary to, a previous treatise already published and entitled “Thermal and Mechanical Aspect of Entropy-Exergy Relationship” [1]. The purpose is to further extend the perspective already adopted to provide an overarching generalization to include chemical systems and phenomena: in particular, an extension to biological molecules, and molecular aggregates, represents the basis to demonstrate the rigorous and reliable analysis of the relationship between entropy and exergy properties and their applications to chemical non-living and living systems. The interest of such an extension relies in the fact that design and experimental analyses and verifications in different fields of application require implementation of extrema principles based on entropy and exergy as non-conservative and additive state properties. Indeed, in non-equilibrium phenomena, maximum or minimum entropy generation (at macroscopic level) or production (at microscopic level) constitute a methodological paradigm implied in the exergy property founded on the very entropy–exergy relationship. Though, exergy property provides a more complete evaluation of processes since it accounts for: (i) reversible non-dissipative conversions among different forms of energy; and (ii) irreversible dissipative conversions determining entropy creation and related exergy destruction. This dual meaning completeness of the exergy property suggests the research of extrema principles in terms of maximum or minimum exergy disgregation (at macroscopic level) and maximum or minimum exergy degradation (at microscopic level). This extension in turn requests a generalization of properties and processes to chemical internal energy in addition to thermal internal energy usually focused to provide demonstrations and applications of exergy property definitions and exergy method applications. Specific reference is made to the school of thought developed at MIT and reported in publications, textbooks and papers, duly mentioned to describe the paradigm of the methodology as well as the conceptual framework of thermodynamics foundations [2].

As it is used in many different contexts and dissertations, it is worth clarifying that the term “generalized” is here used to refer to thermal, chemical and mechanical (and electro-magnetic) aspects of systems and phenomena pertaining to the domain of physical, chemical and biological foundations and applications [3,4].

Moreover, it is worth positing a caveat relating to the concept of heat, mass and work interactions characterizing processes among systems. Indeed, heat, mass and work represent thermal, chemical and mechanical energy transfer and the use of this concept introduces a logical loop in the definition of thermodynamic properties that has been overcome by means of a different set of definitions, assumptions and theorems [5,6]. Despite the use of terms such as “heat or mass or work interaction” should be avoided for this reason, though it is adopted here only to address thermal, chemical and mechanical energy flows and exchanges among systems. In particular, the mass interaction is the homology of heat interaction, whereby particles’ potential energy, in terms of chemical potential, is transmitted between two interacting systems, instead of particles’ kinetic energy transmitted in the form of heat interaction. In finite terms, the mass interaction occurs through mass entering and exiting the system at constant overall mass, implying that chemical potential is the driving force moving chemical energy associated to chemical entropy fluxes. Nevertheless, mass interaction can be obtained with no bulk-flow through the system and by means of stereochemical variations characterized by isomerization of molecules and polymers.

The interest in developing exergy property and the exergetic method has been highlighted in different domains, spreading from industry, ecology, biology, as reported in the literature [7]. In exergo-economic applications, exergy has even become the central quantity of a theory of exergetic cost [8].

## 2. Assumptions and Methods

The dualism consisting of the chemical and mechanical aspects of thermodynamic systems and phenomena represents the chemical–mechanical perspective complementary to the thermal–mechanical one. This conceptual symmetry is further analyzed to provide a definition of the components of entropy and exergy properties relating to mass interactions, typically characterizing chemical processes and chemical internal energy transfer, and work interactions, always occurring along the interaction of any system with a thermal–chemical–mechanical reservoir. Again, in this framework the correlation of chemical potential μ (corresponding to temperature) with respect to chemical internal energy $U^{C}$, and the correlation of pressure P with respect to mechanical internal energy $U^{M}$, constitute an axiomatic schema. This very schema allows to achieve an extended definition of chemical exergy determined by both chemical potential and pressure, both accounted for in terms of difference with respect to the stable equilibrium state of the external reference system (reservoir) state, is considered. A reservoir is posited to be characterized as behaving at constant chemical potentials and constant pressure (in addition to constant temperature) moving along stable equilibrium states [2].

## 3. Chemical and Mechanical Components of Entropy Property

As a logical implication of the second law, stated in terms of existence and uniqueness of the stable equilibrium state of a system, the definition of entropy property is proved by using the non-existence of perpetual motion machines of the second kind (PMM2) [2]; the definition of entropy is expressed through the difference between energy and available energy, or exergy, times a parameter characterizing the reservoir [2]. This inferential method is valid for both thermal and chemical components contributing to the entropy balance of any system in any state. Hence, chemical entropy $S^{C}$, in addition to thermal entropy $S^{T}$, constitutes a property determining the overall internal energy content according to the Euler relation as reported in the literature [2]:(1)$$U=U^{T}+U^{C}+U^{M}=TS+\sum_{i=1}^{r}\mu_{i}n_{i}-PV$$
where $\mu_{i}$ and $n_{i}$ are the chemical potential and the number of moles, respectively, of the *i*-th chemical species for substances composed by r chemical species; this relation, applicable to closed or open systems, hence accounting for “permanent” internal system (non-flow) or “transit” external system (bulk-flow) interactions, can be used to argue for a confrontation between two canonical thermodynamic processes, namely, isothermal and isopotential, as described hereafter. The validity of the phase rule is duly considered since it governs the number of independent intensive quantities determining the thermodynamic state of any system: F=C−P+2, where F is the degree of freedom, C is the number of components, or chemical constituents, and P is the number of phases (solid, liquid, vapor, gas).

In case of isothermal reversible or irreversible processes, the temperature is assumed to remain constant while the system undergoes heat interactions and work interactions simultaneously so that δQ=δW⇒dU=0; in the general case of systems undergoing physical operations or chemical reactions, the chemical potential may change along the isothermal process; this is the case of physical operations, such as phase changes (liquid-to-vapor evaporation of vapor-to-liquid condensation), or direct and inverse chemical or stereochemical reactions in which constitutional, conformational or configurational molecular changes occur. In all those different types of isothermal processes, the only result is that heat interaction is transformed into work interaction, or vice versa; hence, in general, the system undergoes chemical potential variations, even though no mass interactions occur and contribute, with interactions intensity and system density, to determine the pressure of the system in addition to the temperature that, instead, remains constant as assumed to characterize the process.

In cases where an isopotential reversible or irreversible process of open systems is analyzed, chemical potentials are assumed to remain constant within the internal system. Notwithstanding both physical operations or chemical reactions may occur, the system undergoes mass interaction and work interaction simultaneously so that δM=δW⇒dU=0. Indeed, high chemical potential mass input is compensated for by low chemical potential mass output to ensure no variation of the total mass constituting the system and no variation of chemical potential while a portion of input mass chemical potential is transformed into pressure to allow work interaction. Along an isopotential process, the system may undergo temperature variations (e.g., due to compression or expansion of vapors or gases) even though no heat interactions occur; the temperature contributes towards determining the internal pressure of the system; on this basis, the internal energy variation is formulated by means of the total differential of Euler relation and is expressed, in the specific case of isopotential processes, in the following terms:(2)$$dU=dU^{C}+dU^{M}=d\left(\sum_{i=1}^{r}\mu_{i}n_{i}\right)-d\left(PV\right)=\delta M+\delta W=0$$
as far as the mechanical term appearing in this relation is concerned, it is null because the isopotential process at constant mass implies that the mechanical internal energy of the whole system remains constant:(3)−d(PV)=−PdV−VdP=0
where, for the general case of an open system undergoing an isopotential process, the equality PdV=−VdP applies.

Without limiting the generality of this approach, the system considered can be an ideal gas and the thermal form of state equation $PV=\bar{R}T$ applies; though, considering the chemical aspect of the internal system, reference can be made to the chemical form of the state equation that is expressed by means of the chemical potential in lieu of temperature [3]; hence the chemical form of state equation $PV=\bar{R}\mu$ [9] is used to infer that, at constant chemical potential $dU^{M}=-d\left(PV\right)=0$ as a consequence of the definition of isopotential process, and Equation (2) becomes:(4)$$dU=dU^{C}=d\left(\sum_{i=1}^{r}\mu_{i}n_{i}\right)=0$$

The chemical energy can also be expressed by means of a formulation relating to the chemical potential and the chemical entropy in the same form of thermal energy, that is:(5)$$\sum_{i=1}^{r}\mu_{i}n_{i}=\sum_{i=1}^{r}\mu_{i}S_{i}^{C}$$

Besides an equivalence factor, the chemical entropy is directly related to the number of molecules, or moles, of any chemical species constituting the internal system; hence, the total differential is:(6)$$d\left(\sum_{i=1}^{r}\mu_{i}S_{i}^{C}\right)=\sum_{i=1}^{r}\mu_{i}dS_{i}^{C}+\sum_{i=1}^{r}S_{i}^{C}d\mu_{i}$$

However, as the process is isopotential, thus behaving at constant $\mu_{i}$, then the above equation becomes:(7)$$dU=dU^{C}=\sum_{i=1}^{r}\mu_{i}dS_{i}^{C}=0\;\mathrm{that}\;\mathrm{requires}\;dS_{i}^{C}=0$$

The mass interaction occurring along an isopotential process can be realized by means of the addition or subtraction of mass determining the total mass variation of the internal system under consideration; in such a process, both physical operations and chemical reactions are allowed to occur: $dS_{i}^{C}=0$ implies $dn_{i}^{}=0$ that is valid if, and only if, the total mass remains constant but, on the other side, the total mass itself has to change due to mass interaction characterizing the assumed isopotential process; hence, the total mass should remain constant and should change at the same time along the same isopotential process, thus representing an apparent contradiction. The resolution of this contradiction relies in the physical meaning of chemical entropy that, instead, is to be considered as a total entropy (of chemical origin), including chemical and mechanical contributions due to mass interaction related to chemical potential, and work interaction related to pressure.

In this regard, as far as the mechanical aspect of the isopotential process is concerned, a further argument relates to the adiabatic reversible process (non-heat and non-mass interactions with external system) that, hence, is accomplished at constant chemical entropy and constant thermal entropy while chemical potential and pressure change along the process; according to the following equations:(8)$$\begin{array}{c} S^{C}\left(\mu ,V\right)-S_{0}^{C}=C_{n}\ln\frac{\mu }{\mu_{0}}+\bar{R}\ln\frac{V}{V_{0}}\\ S^{C}\left(\mu ,P\right)-S_{0}^{C}=C_{P}\ln\frac{\mu }{\mu_{0}}-\bar{R}\ln\frac{P}{P_{0}} \end{array}$$

The above expressions are obtained from the homologous ones depending on temperatures of the system; the first term of the second member relates to the chemical potential variation due to chemical reactions occurring in the internal system (with inter-particle potential energy variation and no inter-particle kinetic energy variation), and the second term of the second member relates to the mechanical potential, that is to say, pressure variation due to (internal) work interaction; hence, it can be inferred that the chemical entropy variation, associated to mass interaction, is null by definition of non-mass interaction process; therefore, the way chemical entropy remains constant is because of a compensation effect due to the combination of increasing chemical potential and decreasing pressure, or vice versa, in the internal system.

The specific case of an isopotential reversible or irreversible process is typically representative of a system interaction at constant chemical internal energy. This process requires that both chemical internal energy and chemical entropy remain constant since a mass-to-work or work-to-mass conversion occurs isopotentially by definition, i.e., at constant chemical potential (and constant or variable temperature). This operation is determined by equal quantities of mass input and work output, or work input and mass output. Nevertheless, the mass input or output is associated with a transfer of chemical entropy between internal and external system: hence, a transfer of entropy under “chemical” form requires an entropy transformation into “mechanical” form in order to close the total balance of entropy components to zero, as required by Equation (7), reported here again: $dU=dU^{C}=\sum_{i=1}^{r}\mu_{i}dS_{i}^{C}=0$ that implies $dS_{i}^{C}=0$; as mechanical internal energy does not undergo any variation as assumed, this mechanical form of entropy is correlated to work output or input and is determined by pressure and volume variations. In this regard, mechanical entropy is an additional component, and is consistent with, the canonical formulation of entropy calculated for any process, including isopotential, implying that chemical entropy is determined solely by the mass interaction. This conclusion is in compliance with the result provided with the same rationale as for an isothermal reversible process and is described in the process reported in the homologous procedure already mentioned [1].

The analysis described above demonstrates that entropy, in its more general meaning and characterizing internal energy, is constituted by two different and independent components; the first one is the “chemical entropy” that remains constant along an adiabatic reversible process (usually termed as isoentropic), where, instead, only work interaction occurs; the second one is the “mechanical entropy” that remains constant along an isovolumic reversible process where mass interaction (or heat interaction) only occurs. In addition, it can be posited that entropy property S, appearing in the expression of internal energy $U=TS+\sum_{i=1}^{r}\mu_{i}n_{i}-PV$, specifically represents the thermal component, or thermal entropy, out of the overall contribution that, nevertheless, remains consistent with, and does not disprove, the above analysis. From a methodological standpoint, the relationship between entropy and exergy properties represents the basis for assuming and proving that chemical and mechanical components set forth for entropy remain valid for chemical exergy and mechanical exergy, respectively.

## 4. Chemical Exergy Derived from Carnot Chemical Direct Cycle

The definition of chemical exergy analyzed here, among others reported in the literature [10], is based on mass and work interactions and addresses the chemical aspect as a symmetric concept with respect to thermal aspects in the consideration of internal energy contributions. In terms of interactions with the reservoir, the chemical exergy is formulated as the maximum theoretical net useful work withdrawn as a portion of the internal energy of the system, constituting the available energy, along a process leading the system-reservoir composite to the stable equilibrium state. This useful work is calculated on the basis of thermodynamic efficiency of the Carnot chemical direct cycle operating between the variable chemical potential μ of a system A, and the constant chemical potential $\mu_{R}$ of a reservoir R assumed as the external reference system:(9)$$dEX_{}^{C}=\delta W_{REV}^{NET}=\delta W_{REV}^{CONVER}+\delta W_{REV}^{TRANSF}$$
where the differential form of chemical exergy is expressed by means of the sum of two terms: (i) a first contribution $\delta W_{REV}^{CONVER}$ deriving from the conversion of mass interaction into work interaction through a mass-to-work Carnot chemical direct cyclic process [11,12] converting the chemical energy, available at higher chemical potential $\mu_{}^{HC}$, by means of an ideal cyclic machinery operating between $\mu_{}^{HC}$ and the reservoir at $\mu_{R}^{LC}$; (ii) a second contribution $\delta W_{REV}^{TRANSF}$ deriving from the transfer of mechanical energy by means of work interaction through a cyclic process resulting from system volume variation by means of an ideal machinery operating between $P_{}^{HP}$ and the reservoir at $P_{R}^{LP}$; for sake of generality, mass and work interactions can occur either sequentially in different processes or concurrently within the same process; both result from the generalized available energy of a simple system as defined in the approach by Gyftopoulos and Beretta [2].

The rationale to define chemical exergy is based on the confrontation of thermal and chemical aspect of cyclic processes. Usually, temperature is the intensive property determining the Carnot cycle representing the highest efficiency cyclic process and constituting the consequence of the non-existence of perpetual motion machine of the second kind (PMM2) [13]. However, if the same Carnot cycle is regarded as characterized by the chemical potential as an intensive property, instead of temperature, then the Carnot chemical cycle constitutes the symmetric process of a Carnot thermal cycle, considering pressure as the common reference [13]. Hence, based on the chemical potential, a chemical machine model can be described in terms of a chemical conversion cyclic process as the homology of a thermal conversion cyclic process for which balances and efficiencies can be stated [11,12]. In this sense, the above equation, expressing the chemical exergy in differential terms, can be reformulated in the following form:(10)$$\begin{array}{c} dEX_{}^{C}=\delta W_{REV}^{NET}=\eta_{id}^{CARNOT-CHEMICAL-DIRECT}\cdot \delta M_{}^{HC}+\delta W_{REV}^{TRANSF}\\ =\frac{\delta W}{\delta M_{ISOPOTENTIAL}^{HC}}\cdot \delta M_{}^{HC}-PdV+P_{R}dV\\ =\left(1-\frac{\mu_{R}}{\mu }\right)\cdot \delta M_{}^{HC}+\left(1-\frac{P_{R}}{P}\right)\cdot \delta W_{}^{HP} \end{array}$$
where $\delta M_{ISOPOTENTIAL}^{HC}$ represents the infinitesimal mass interaction along the process at higher chemical potential μ different from the chemical potential $\mu_{R}$ of the reservoir; $\delta M_{}^{HC}$ represents the infinitesimal mass interaction along any process for which chemical exergy is calculated; $\delta W_{}^{HP}$ is the infinitesimal work interaction at (variable) high pressure P alongside the process, higher (or lower) with respect to the reservoir (constant) pressure $P_{R}$; and the two terms in the last member of the above equation are the consequence of the role of pressure corresponding to the role of chemical potential with respect to mass in chemical exergy.

The above equation is similar to the already known canonical definition of physical exergy [14,15,16]; this expression is used to define the exergy that is identified by the superscript “C”, standing for “Chemical”, according to the definition reported in the literature [13] as pointed out above.

In finite terms, considering that $\delta W_{}^{HP}=-PdV$:(11)$$\begin{array}{c} EX^{C}=W_{10}\\ =\int_{0}^{1}\left(1-\frac{\mu_{R}}{\mu }\right)\cdot \delta M^{HC}+\int_{0}^{1}\left(1-\frac{P_{R}}{P}\right)\cdot \delta W^{HP}\\ =M_{10}^{HC}-\mu_{R}\int_{0}^{1}\frac{d\mu^{HC}}{\mu^{HC}}+W_{10}^{HP}+P_{R}\cdot \left(V_{1}-V_{0}\right) \end{array}$$
where $W_{10}$ is the maximum theoretical net useful work output extracted from the generalized available energy as results from the interaction between system and reservoir; $M_{10}^{HC}$ is the mass interaction alongside the process from the higher isopotential process at μ to the lower isopotential process at $\mu_{R}$ (as a particular case, mass interaction can occur alongside an isopotential process); and $W_{10}^{HP}$ is again the work interaction from the higher isopotential process at μ to the lower isopotential process at $\mu_{R}$. This equation expresses the chemical exergy $EX^{C}$ in finite terms as the sum of contributions deriving from cyclic processes where the first one is a mass-to-work ideal cyclic conversion and the second one is an HP-work-to-LP-work ideal cyclic transformation.

The sum of $M_{10}^{HC}$ and $W_{10}^{HP}$ can also be expressed by integrating the Equation (2):(12)$$M_{10}^{HC}+W_{10}^{HP}=U_{1}-U_{0}=M_{V}\cdot \left(\mu_{1}-\mu_{0}\right)$$
where the equivalence represents the amount of mass interaction only in the isovolumic processes connecting two states at different chemical potentials. Therefore, chemical exergy can also be associated to sequential isovolumic-isopotential processes connecting any state 1 with a different state 0 of the system. The integral operation results in the expression of chemical exergy, Equation (11). In infinitesimal terms, it constitutes the definition of entropy according to the canonical formulation or, as here proposed, the chemical component of entropy property identified by the superscript “chemical”; the expression in finite terms becomes:(13)$$EX^{C}=W_{10}=\left(U_{1}-U_{0}\right)-\mu_{R}\cdot \left(S_{1}^{C}-S_{0}^{C}\right)+P_{R}\cdot \left(V_{1}-V_{0}\right)$$
where the system-reservoir composite interaction at constant chemical potential $\mu_{R}$ and constant pressure $P_{R}$ of the reservoir results. This formulation does not contradict the homologous one, proposed by Gyftopoulos and Beretta, deduced from the definition of generalized available energy with respect to an external reference system at constant chemical potential $\mu_{R}$ and constant pressure $P_{R}$ behaving as a reference external system.

## 5. Mechanical Exergy Derived from Carnot Chemical Inverse Cycle

The correlation between chemical entropy and chemical exergy clarified in the previous sections is the basis to analyze the entropy–exergy relationship. This analysis is carried out starting from a mechanical standpoint to develop the concept of exergy related, in this case, to work and pressure. To do so, the existence of the mechanical component of entropy already proved is taken into consideration. This different standpoint is viable because the equality of pressure between system and reservoir is an additional condition of the existence and uniqueness of stable equilibrium states of the system-reservoir composite, other than the equality of chemical potential. Indeed, both pressure and chemical potential are thermodynamic potentials driving any equilibrium or non-equilibrium process in the direction of stable equilibrium.

The definition of exergy formulated by the Carnot chemical direct cycle consists of chemical exergy which highlights the role of chemical potential in mass-to-work conversions. On this basis, the research for a definition of mechanical exergy expressed by the inverse cycle is the logical consequence. The objective becomes the physical meaning of the pressure in the opposite process, that is work-to-mass conversion. For the mechanical standpoint too, the general formulation of exergy, in infinitesimal terms, derives from the relationship founded on the Carnot chemical (inverse) cycle and the related expression of thermodynamic efficiency determined by chemical potentials of system and reservoir.

As far as the Carnot cycle is concerned, the usual expression of its performance in terms of thermodynamic efficiency is related to high temperature and low temperature isothermal processes through heat interactions with two reservoirs. Though, the thermodynamic potential constituted by the temperature, or by the inter-particle kinetic energy within the internal system, is continuously transformed into inter-particle potential energy constituting the chemical potential of molecules. In turn, the chemical potential constitutes a thermodynamic potential determining the performance of such a chemical cyclic process. Focusing the performance of chemical cyclic process, it is expressed by means of homologous expression as thermal cyclic processes. Hence, if reference is made to high and low chemical potentials defined as $\mu^{HC}$ and $\mu^{LC}$ characterizing isopotential processes of the “Carnot chemical cycle”, then the formulation of ideal cycle efficiency $\eta_{id}^{C}$ is stated as:(14)$$\eta_{id}^{C}=1-\frac{\mu^{LC}}{\mu^{HC}}$$

These isopotential processes are intended to be characterized by mass interaction input and work interaction output, and vice versa, while the chemical potential of the mass constituting the system remains constant: this means that entering mass implies reducing chemical potential due to chemical reactions occurring at constant temperature while work is exiting the system.

The chemical Carnot cycle considered here will be used to define chemical exergy on the basis of its homology with the canonical thermal Carnot cycle usually referred to in the literature. This chemical cycle, elaborated through ideal processes, is symmetric because it consists of four elaborations, each pair of which is of the same type (isodiabatic), as represented in Figure 1. In case the operating internal system is a perfect gas as assumed, the alternating polytropic processes (two adiabatic and two isopotential), behave according to the following property:(15)$$\frac{V_{1}}{V_{0}}=\frac{V_{1C}}{V_{0C}}\;;\;\frac{P_{1}}{P_{0}}=\frac{P_{1C}}{P_{0C}}\;;\;\frac{\mu_{1}}{\mu_{0}}=\frac{\mu_{1C}}{\mu_{0C}}$$
where the meaning of these ratios is that properties at the end of isodiabatic processes are proportional, therefore the amount of work interaction between internal and external system is the same for both adiabatic compression from 0 to 1 and expansion from 1C to 0C processes; this amount of work interaction is calculated by means of the following expression:(16)$$W=\frac{1}{K-1}P_{0}V_{0}\left[{\left(\frac{P_{1}}{P_{0}}\right)}^{\frac{K-1}{K}}-1\right]$$
where this depends on the equality $\frac{P_{1}}{P_{0}}=\frac{P_{1C}}{P_{0C}}$ and therefore input and output W (with different sign) is equal for the two adiabatic reversible processes. The resulting work interaction balance along the whole cycle accounts for the algebraic sum of work interaction contributions due to both isopotential processes only where mass and work interactions are exchanged simultaneously in directly proportional and equal amounts. This property enables expression of the thermodynamic efficiency of the Carnot chemical cycle of an open bulk-flow system both in terms of mass interaction or work interaction. That thermodynamic efficiency can be expressed either in terms of mass interaction only or in terms of work interaction only due to the equality of mass-work input-output, or vice versa, alongside the isopotential processes as represented in Figure 1:(17)$$\eta_{id}^{C-DIR}=\frac{W}{M^{HC}}=\frac{W^{HP}-W^{LP}}{M^{HC}}=\frac{W^{HP}-W^{LP}}{W^{HP}}=\frac{M^{HC}-M^{LC}}{M^{HC}}$$

As far as the inverse cycle is concerned, if the role of used mass interaction at high chemical potential $M^{HC}$ and utilized total work interaction $M^{HC}$ are replaced by used work interaction $W^{HP}$ and utilized total mass interaction M, the following expression applies:(18)$$\eta_{id}^{C-INV}=\frac{M}{W^{HP}}=\frac{M^{HC}-M^{LC}}{W^{HP}}=\frac{M^{HC}-M^{LC}}{M^{HC}}=\frac{W^{HP}-W^{LP}}{W^{HP}}=\eta_{id}^{C-DIR}$$
where this equation, for a Carnot chemical inverse cycle, is obtained by assuming the meaning of used and utilized interactions with proper input and output: used mass $M^{HC}$ in the direct cycle corresponds to the used work $W^{HP}$ in the inverse cycle; moreover, the utilized total work W in the direct cycle corresponds to the utilized total mass M in the inverse cycle; as a consequence, the efficiency of a Carnot chemical direct cycle, depending on isopotential processes only, remains unchanged if the Carnot chemical inverse cycle is considered with the corresponding opposite processes; hence the following equality is demonstrated:(19)$$\eta_{id}^{C-INV}=\frac{M}{W^{HP}}=\frac{W}{W^{HC}}=1-\frac{\mu^{LC}}{\mu^{HC}}=\eta_{id}^{C-DIR}$$

It is noteworthy that this approach focuses the concept of exergy and its definitions in terms of used and utilized quantities; thus, it is different from the concept of coefficient of performance (CoP) adopted for refrigeration and cryogenic processes for which used and utilized flows are different and in compliance with operative performances in applications in machinery and plants.

The definition of exergy based on the direct cycle as chemical exergy which is determined by chemical potential in mass-to-work conversion, can be complemented by the symmetric definition of mechanical exergy founded on the inverse cycle; in this case, the physical meaning of pressure in the opposite work-to-mass conversion, determines the pressure level of work interactions alongside the higher chemical potential, and higher pressure, isopotential processes of the Carnot chemical inverse cycle.

The concept of equivalence and interconvertibility, demonstrated by Gaggioli [14,15,16], can be stated in different terms: “useful work is not better than useful mass, and the available energy results in maximum net useful mass or, equivalently, maximum net useful work, or the combination of both.” Thus, the definition of mechanical exergy representing, in this case, the maximum net useful mass withdrawable from the available energy, in infinitesimal terms, can be expressed as:(20)$$dEX^{M}=\delta M_{REV}^{NET}=\delta M_{REV}^{CONVER}+\delta M_{REV}^{TRANSF}$$
where the first term of the last member $\delta M_{REV}^{CONVER}$ is the net amount of mass interaction resulting from the balance of a Carnot chemical inverse cycle converting the available work at pressure P into mass through the interaction with a reservoir at constant pressure $P_{R}$; the second term of the last member $\delta M_{REV}^{TRANSF}$ is the net amount of available energy transferred from the external to the internal system by means of mass interaction alongside a non-cyclic or cyclic process; mass and work interactions are accounted for occurring either successively or simultaneously, and both derive from the generalized available energy of a simple system as defined by Gyftopoulos and Beretta [2]. Hence, in differential terms:(21)$$dEX^{M}=\delta M_{REV}^{NET}=\eta_{id}^{CHEMICAL-CARNOT-INVERSE}\cdot \delta W^{HP}+\delta M_{REV}^{TRANSF}$$

On the basis of Equation (15):(22)$$\begin{array}{c} dEX^{M}=\frac{\delta M}{\delta W_{ISOTHERMAL}^{HP}}\cdot \delta W^{HP}+\mu dS^{C}-\mu_{R}dS^{C}\\ =\frac{\delta M}{\delta W_{ISOTHERMAL}^{HP}}\cdot \delta W^{HP}+\left(\mu -\mu_{R}\right)dS^{C}\\ =\left(1-\frac{\mu_{R}}{\mu }\right)\cdot \delta W^{HP}+\left(1-\frac{\mu_{R}}{\mu }\right)dM^{HC} \end{array}$$

The formulation of chemical exergy is now reversed to define the mechanical exergy, identified by the superscript “*M*”, that is not related to exergy associated to center-of-mass macroscopic kinetic and potential energy, already termed as “kinetic exergy” and “potential exergy” according to the literature.

After replacing work with mass, the mechanical exergy in finite terms is formulated as:(23)$$\begin{array}{c} EX^{M}=M_{10}=\int_{0}^{1}\left(1-\frac{\mu_{R}}{\mu }\right)\cdot \delta W^{HP}+\int_{0}^{1}\left(\mu -\mu_{R}\right)\cdot dS^{C}\\ =W_{10}^{HP}-\mu_{R}\int_{0}^{1}\frac{\delta W^{HP}}{\mu }+M_{10}^{HC}-\mu_{R}\left(S_{1}^{C}-S_{0}^{C}\right) \end{array}$$
where $M_{10}$ is the maximum theoretical net useful mass output obtained by means of the generalized available energy resulting from the interaction process between system and reservoir; $W_{10}^{HP}$ is the work interaction from higher isopotential curve at μ and corresponds with the state at pressure P to lower isopotential curve at $\mu_{R}$; as a particular case, the work interaction can occur alongside an adiabatic reversible process; the sum of terms $W_{10}^{HP}$ and $M_{10}^{HC}$ in the last member of previous equation can also be expressed as:(24)$$W_{10}^{HP}+M_{10}^{HC}=U_{1}-U_{0}=C_{V}\left(\mu_{1}-\mu_{0}\right)$$

This equation expresses the equivalence with the sole amount of work interaction in a chemical (and thermal) isoentropic process (where work interaction only occurs), between two different chemical potentials. Hence, the mechanical exergy characterizes an isoentropic-isopotential sequential process connecting the generic state 1 with the stable equilibrium state 0 of the system–reservoir composite. If the chemical state equation $PV=\bar{R}\mu$ is adopted and used in the expression of mechanical exergy, then:(25)$$EX^{M}=Q_{10}=\left(U_{1}-U_{0}\right)-\bar{R}\mu_{R}\int_{0}^{1}\frac{\delta W^{HP}}{PV}-\mu_{R}\left(S_{1}^{C}-S_{0}^{C}\right)$$

The integrand term $\frac{\delta W^{HP}}{PV}$ of the above equation is formally homologous of the integrand term $\frac{dM^{HC}}{\mu }$ representing the very definition of chemical entropy according to the concept and the definition of entropy property as per Clausius formulation; on the basis of this formal homology extended to work interaction and the mechanical internal energy, the definition of mechanical entropy is derived and formulated as:(26)$$dS^{M}=\frac{\delta W^{HP}}{PV}$$
where the factor 1/PV is the integrating factor of the infinitesimal work interaction $\delta W^{HP}$ that changes the integrand function into an exact differential function; indeed, assuming the expression of mechanical exergy previously reported as Equation (22), and considering that $\delta W^{HP}=-PdV$ then it is allowed to differently express the mechanical exergy (of chemical origin) as:(27)$$\begin{array}{c} EX^{M}=M_{10}\\ =\left(U_{1}-U_{0}\right)-\bar{R}\mu_{R}\int_{0}^{1}\frac{\delta W^{HP}}{PV}-\mu_{R}\left(S_{1}^{C}-S_{0}^{C}\right)\\ =\left(U_{1}-U_{0}\right)+\mu_{R}\int_{0}^{1}\bar{R}\frac{dV}{V}-\mu_{R}\left(S_{1}^{C}-S_{0}^{C}\right)\\ =\left(U_{1}-U_{0}\right)+\mu_{R}\left(\bar{R}\ln V_{1}-\bar{R}\ln V_{0}\right)-\mu_{R}\left(S_{1}^{C}-S_{0}^{C}\right) \end{array}$$
where it relates to the work interaction with the environmental system represented by the mechanical reservoir; therefore, considering that the chemical state equation $PV=\bar{R}\mu$ applies, then:(28)$$EX^{M}=M_{10}=\left(U_{1}-U_{0}\right)+P_{R}V_{R}\left(\ln V_{1}-\ln V_{0}\right)-\mu_{R}\left(S_{1}^{C}-S_{0}^{C}\right)$$
where the homology with the expression of chemical exergy (and thermal exergy) demonstrates the common origin of all exergy components deriving from conversion processes from one energy form to a different one characterized by entropy variations occurring along those processes; to complete this homology, the integrating factors included in the integration function are similar:(29)$$dS^{T}=\frac{\delta Q^{HT}}{T}\;\mathrm{similar}\;\mathrm{to}\;dS^{C}=\frac{\delta M^{HC}}{\mu }\;\mathrm{similar}\;\mathrm{to}\;dS^{M}=\frac{\bar{R}dV}{V}$$
where the last differential is integrated, the equation in finite terms becomes:(30)$$S^{M}=\bar{R}\ln V+C$$

Hence, $dS^{M}$ being an exact differential function, then $S^{M}$ is a state property depending on the state parameter volume and can be adopted as the formal definition of mechanical entropy; moreover, as volume is additive, the mechanical entropy is an additive property. As far as the dimensional analysis is concerned, since the logarithmic function is dimensionless, then the dimension of mechanical entropy is related to $\bar{R}$ having dimensions $\left(J\cdot {\mathrm{kg}}^{-1}\cdot K^{-1}\right)$ or $\left(J\cdot {\mathrm{mol}}^{-1}\cdot K^{-1}\right)$ that are identical to the chemical entropy and thermal entropy dimensions. In this regard, the relationship between mechanical exergy and volume, and pressure as a consequence, constitutes the rationale for considering the equality of pressure between system and reservoir, as an additional condition of mutual stable equilibrium to be accounted for in the definitions of available energy and exergy, and hence in the definition of entropy property related to, and derived from, energy and available energy or exergy according to the proof method demonstrated and reported in the literature. The physical meaning of mechanical exergy can be ascribed to the combination of pressure characterizing the mechanical internal energy of the system, and the pressure of work interaction occurring between system and reservoir. It is noteworthy that the demonstration procedure described here is, in its rationale, identical to the one stated to achieve the mechanical entropy definition based on thermal entropy using the corresponding quantities to replicate the proof.

## 6. Generalized Chemical Exergy Related to Chemical-Mechanical Reservoir

The definition of chemical entropy and mechanical entropy, derived and expressed from chemical exergy and mechanical exergy, respectively, is accounted for here to generalize the conceptual definition of chemical exergy including mass interaction, in addition to work interaction, characterizing interaction processes occurring between system and reservoir. On the basis of equivalence and interconvertibility proposed by Gaggioli et al. [9,10] for thermal and mechanical aspect of interactions, and here mutuated for chemical and mechanical interactions, the exergy of a system interacting with a reservoir results in the following statements:(1)Exergy is the available work or maximum theoretical net useful work constituting the chemical exergy;(2)Exergy is the available mass or maximum theoretical net useful mass constituting the mechanical exergy;

The generalization of chemical exergy proposed here is, for the above rationale, implicated with the chemical exergy underpinned by the Carnot chemical direct cycle efficiency and the high chemical potential mass interaction; chemical exergy additionally contributes to the mechanical exergy underpinned by the Carnot chemical inverse cycle efficiency and the high pressure work interaction; both exergies are defined considering a chemical–mechanical reservoir at constant chemical potential and constant pressure behaving at permanent stable equilibrium according to the canonical definition of reservoir. The generalized chemical exergy outlined above takes into account the implication of pressure in work interaction that generates different amounts of mass interaction depending on different pressure values at which the work interaction occurs. In different terms, it can be stated that the same amount of available mechanical internal energy transferred by means of work interaction can be used at different pressure of the system with respect to the constant pressure of the (mechanical) reservoir, to be converted into mass interaction at different chemical potentials. Hence, the useful work, available in the form of mechanical available energy, is also evaluated in terms of the second law by means of the Carnot chemical inverse cycle, producing the mass interaction output: therefore, mass-to-work conversion and work-to-mass conversions are accounted for simultaneously—this implies that the generalized chemical exergy can be regarded in the perspective of an “exergy of exergy” that makes work interaction equivalent to, and interconvertible with, mass interaction, and vice versa.

Before achieving the formulation of the generalized chemical exergy, the differential form of internal energy in differential terms according to Gibbs’ equation is considered:(31)$$dU=\sum_{i=1}^{r}\mu_{i}dn_{i}-PdV=\delta M+\delta W$$

This can be reformulated in different terms by adopting the chemical entropy and the mechanical entropy previously defined and specified for all chemical substances constituting the internal system; this reformulation is a crucial step in the direction of a generalized Gibbs equation that, in this perspective, is modified into the following:(32)$$dU=\sum_{i=1}^{r}\mu_{i}dS_{i}^{C}-\sum_{i=1}^{r}\frac{P_{i}V_{i}}{\bar{R}}dS_{i}^{M}=\delta M+\delta W$$
where, in turn, it can be expressed by means of the chemical state Equation [5]:(33)$$\begin{array}{c} dU=\sum_{i=1}^{r}\mu_{i}dS_{i}^{C}-\sum_{i=1}^{r}\mu_{i}dS_{i}^{M}\\ =\sum_{i=1}^{r}\mu_{i}\left(dS_{i}^{C}-dS_{i}^{M}\right)=\delta M+\delta W \end{array}$$

The term $\left(dS_{i}^{C}-dS_{i}^{M}\right)$ represents the differential generalized entropy $dS_{i}^{G}$ which, in finite terms, is $S_{i}^{G}=S_{i}^{C}-S_{i}^{M}$ associated to, and depending on, the chemical potentials and is determined by mass interaction and work interaction contributing to the variation of the internal energy. Equation (33) above can be expressed as:(34)$$dU=\sum_{i=1}^{r}\mu_{i}dS_{i}^{G}=\delta M+\delta W$$
where, in finite terms, U being a state property determined by two independent variables, the following generalized Gibbs equation is deduced:(35)$$U=U\left(S,V\right)=\sum_{i=1}^{r}\mu_{i}\Delta S^{G}=M+W$$

The generalized entropy is the result of the contribution of chemical and mechanical components and represents the rationale for resolving the apparent inconsistency expressed by the statement: $dU=dU^{C}=\sum_{i=1}^{r}\mu_{i}dS_{i}^{C}=0$ implying that $dS_{i}^{C}=0$; indeed, the Gibbs equation is allowed to be null because of the two terms of $S_{i}^{G}=S_{i}^{C}-S_{i}^{M},$ which, in the special case of an isopotential process of a perfect and single-phase homogeneous gas describing the internal system, are expressed as:(36)$$\Delta S_{ISOPOTENTIAL}^{C}=C_{n}\ln\frac{\mu }{\mu_{0}}+\bar{R}\ln\frac{V}{V_{0}}$$
(37)$$\Delta S_{ISOPOTENTIAL}^{M}=\bar{R}\ln\frac{V}{V_{0}}$$

These two terms used to replace the corresponding ones in the $S_{i}^{G}=S_{i}^{C}-S_{i}^{M}$ become:(38)$$\Delta S_{ISOPOTENTIAL}^{G}=\Delta S_{ISOPOTENTIAL}^{C}-\Delta S_{ISOPOTENTIAL}^{M}=C_{n}\ln\frac{\mu }{\mu_{0}}=0$$
confirming that ΔU=0 for an isopotential reversible or irreversible process as $\Delta S_{ISOPOTENTIAL}^{G}=0$ as required to resolve the inconsistency of conditions $dU=dU^{C}=\sum_{i=1}^{r}\mu_{i}dS_{i}^{C}=0$ implying $dS_{i}^{C}=0$ before positing.

As far as isovolumic processes are concerned, the same approach is applied by evaluating the two components of generalized entropy along the process:(39)$$\Delta S_{ISOVOLUMIC}^{C}=C_{n}\ln\frac{\mu }{\mu_{0}}$$
(40)$$\Delta S_{ISOVOLUMIC}^{M}=0$$

Thus, the sum of the two contributions is:(41)$$\Delta S_{ISOVOLUMIC}^{G}=\Delta S_{ISOVOLUMIC}^{C}-\Delta S_{ISOVOLUMIC}^{M}=C_{n}\ln\frac{\mu }{\mu_{0}}$$

Therefore, the generalized entropy is identical to the chemical entropy, that confirming the dependence on the chemical potential as the overall and unique thermodynamic potential determining the state of the system.

In case of an isobaric process, the following applies:(42)$$\Delta S_{ISOBARIC}^{C}=C_{P}\ln\frac{\mu }{\mu_{0}}-\bar{R}\ln\frac{P}{P_{0}}$$
(43)$$\Delta S_{ISOBARIC}^{M}=\bar{R}\ln\frac{V}{V_{0}}$$

Again, the sum of the two contributions is:(44)$$\begin{array}{c} \Delta S_{ISOBARIC}^{G}=\Delta S_{ISOBARIC}^{C}-\Delta S_{ISOBARIC}^{M}\\ =C_{P}\ln\frac{\mu }{\mu_{0}}-\bar{R}\ln\frac{P}{P_{0}}-\bar{R}\ln\frac{V}{V_{0}}\\ =C_{n}\ln\frac{\mu }{\mu_{0}}+\bar{R}\ln\frac{V}{V_{0}}-\bar{R}\ln\frac{V}{V_{0}}\\ =C_{n}\ln\frac{\mu }{\mu_{0}} \end{array}$$

Finally, for an adiabatic reversible process:(45)$$\Delta S_{ADIABATIC}^{C}=0$$
(46)$$\Delta S_{ADIABATIC}^{M}=\bar{R}\ln\frac{V}{V_{0}}$$
(47)$$\begin{array}{c} \Delta S_{ADIABATIC}^{G}=\Delta S_{ADIABATIC}^{C}-\Delta S_{ADIABATIC}^{M}=0-\Delta S_{ADIABATIC}^{M}\\ =-\bar{R}\ln\frac{V}{V_{0}}=C_{n}\ln\frac{\mu }{\mu_{0}} \end{array}$$
hence demonstrating, by means of Equations (8), the existence of the relationship between pressure, that changes with volume, and the generalized entropy in the special case of absence of mass interaction determining chemical entropy null variations.

To summarize, a first outcome is that the method applied to explain the mechanical entropy contribution has led to resolve the apparent controversy already mentioned and provides a formal definition of mechanical entropy related to the pressure, with a direct implication with the definition of mechanical exergy property. A second outcome, deriving from the above method, concerns the dependence of the generalized chemical entropy solely on the chemical potential in all thermodynamic processes analyzed above; this outcome can be derived from the physical meaning of internal energy pertaining to a real, multi-phase, non-homogeneous, internal system characterized by atomic-molecular chemical bonds and interactions regardless of the thermal state and heat interactions between internal and external systems.

A caveat concerning the assumption that the canonical processes above are not limited to reversible conditions, irreversible processes are accounted for.

That said, on the basis of the relationship between generalized entropy and internal energy, if the external system behaves as a chemical and mechanical reservoir in stable equilibrium state at constant chemical potential and pressure, the internal energy balance of the system-reservoir composite is expressed as:(48)$$\begin{array}{c} EX^{C}=-\left(W^{AR\leftarrow }\right)-\left(M^{AR\leftarrow }\right)\\ =\Delta U^{SYSTEM}+\Delta U^{RESERVOIR}\\ ={\Delta U}_{W}^{SYSTEM}+\Delta U^{R,W}+{\Delta U}_{M}^{SYSTEM}+\Delta U^{R,M} \end{array}$$

The conceptual meaning of this expression is that ${\Delta U}_{W}^{SYSTEM}+\Delta U^{R,W}$ equals the mechanical exergy converted into chemical exergy, and ${\Delta U}_{M}^{SYSTEM}+\Delta U^{R,M}$ equals the chemical exergy converted into mechanical exergy; in different terms:(49)$$EX^{C}=-\left(W^{AR\leftarrow }\right)-\left(M^{AR\leftarrow }\right)=\left(U-U_{0}\right)-M_{R}-W_{R}$$
where $M_{R}$ is the minimum mass interaction representing the (minimum) mechanical exergy (Equation (28)) lost to the chemical reservoir and $W_{R}$ is the minimum work interaction representing the (minimum) chemical exergy (Equation (13)) lost to the mechanical reservoir. The symbol $EX^{C}$ (or, according to some authors, $X^{C}$), in lieu of M and W, is adopted here to identify the chemical exergy generalized in its physical and chemical meaning as deriving from the combination of useful work and useful mass. The arrow in the superscript means that the interaction enters the system, according to the symbology adopted by Gyftopoulos and Beretta [2].

The Carnot cycle represented in Figure 1 constitutes the rationale for the generalized formulation of chemical exergy; indeed, the isopotential process verifies the equality $M^{AR}=W^{AR}$ alongside both high and low chemical potential processes where, instead, the chemical potential is constant but the pressure is not; therefore, $W^{AR}$ at decreasing pressure constitutes an amount of (chemical) exergy that should be considered a loss of mechanical internal energy since it is released isopotentially to the reservoir while chemical internal energy is transferred from the reservoir to the system at stable equilibrium conditions; those isopotential processes are the result of chemical-to-mechanical and mechanical-to-chemical internal energy transformations implying entropy transformation appearing in the equation of generalized chemical exergy:(50)$$\begin{array}{c} EX^{G}=-\left(W^{AR\leftarrow }\right)-\left(M^{AR\leftarrow }\right)=\Delta U^{SYSTEM}+\Delta U^{RESERVOIR}\\ =\left(U_{1}-U_{0}\right)\;\;\;\mathrm{variation}\;\mathrm{of}\;\mathrm{internal}\;\mathrm{energy}\;\mathrm{of}\;\mathrm{the}\;\mathrm{system}\\ -\sum_{i=1}^{r}\mu_{i}\Delta S_{i}^{C}\;\;\;\mathrm{energy}\;\mathrm{conversion}\;\mathrm{within}\;\mathrm{the}\;\mathrm{system}\\ -\sum_{i=1}^{r}\mu_{i}\Delta S_{i}^{C,R}\;\;\mathrm{chemical}\;\mathrm{energy}\;\mathrm{transfer}\;\mathrm{between}\;\mathrm{system}\;\mathrm{and}\;\mathrm{reservoir}\\ +P_{R}\Delta V^{R}\;\mathrm{mechanical}\;\mathrm{energy}\;\mathrm{transfer}\;\mathrm{system}-\mathrm{to}-\mathrm{reservoir} \end{array}$$

It is of crucial importance highlighting that the concept of entropy conversion is inherent to the concept of energy conversion occurring in any cyclic process, and, for this very reason, intrinsic to the concept of exergy; hence, entropy conversion occurring along a cyclic process implies the additional term expressing the contribution of the mechanical component to the overall cycle entropy balance and the subsequent exergy balance representing the basis of a property’s efficiency and, finally, the performance quantification. Replacing the expressions of chemical exergy $EX^{C}$ and mechanical exergy $EX^{M}$ in the above equation of generalized exergy $EX^{G}$, the following equation is derived:(51)$$\begin{array}{c} EX^{G}=\left(U_{1}-U_{0}\right)\\ -\sum_{i=1}^{r}\mu_{i,R}\Delta S_{i}^{C}\;\;\;\;\;\mathrm{chemical}\;\mathrm{energy}\;\mathrm{conversion}\;\mathrm{loss}\;\mathrm{released}\;\mathrm{to}\;\mathrm{reservoir}\\ +P_{R}\Delta V_{R}\;\;\;\;\;\;\;\mathrm{mechanical}\;\mathrm{energy}\;\mathrm{transformation}\;\mathrm{loss}\;\mathrm{released}\;\mathrm{to}\;\mathrm{reservoir}\\ +P_{R}V_{R}\left(\ln V_{1}-\ln V_{0}\right)\;\;\;\mathrm{mechanical}\;\mathrm{energy}\;\mathrm{conversion}\;\mathrm{loss}\;\mathrm{released}\;\mathrm{to}\;\mathrm{reservoir}\\ -\sum_{i=1}^{r}\mu_{i,R}\Delta S_{i}^{C}\;\;\;\;\;\mathrm{chemical}\;\mathrm{energy}\;\mathrm{transformation}\;\mathrm{loss}\;\mathrm{released}\;\mathrm{to}\;\mathrm{reservoir} \end{array}$$

The term $P_{R}V_{R}\left(\ln V_{1}-\ln V_{0}\right)$ constitutes the “entropic-mechanical” component taking into account the entropy conversion undergone by the system along the conversion cycle (and in particular due to adiabatic processes) and representing a contribution, in addition to the chemical entropy, to the overall cycle balance.

As all properties are additive, the generalized chemical exergy can be stated in the following explicit form:$$EX^{G}=-\left(W^{AR\leftarrow }\right)-\left(M^{AR\leftarrow }\right)$$
(52)$$+{\left[\left(U-U_{0}\right)-\sum_{i=1}^{r}\mu_{i,R}\Delta S_{i}^{C}+P_{R}\Delta V_{R}\right]}^{C}$$
$$+{\left[\left(U-U_{0}\right)+P_{R}V_{R}\left(\ln V_{1}-\ln V_{0}\right)-\sum_{i=1}^{r}\mu_{i,R}\Delta S_{i}^{C}\right]}^{M}$$
where: the first term, that is the first square parenthesis (“chemical”) of the second member of Equation (52), is the contribution relating to the variation of internal energy due to the mass interaction corresponding to the chemical exergy; the second term, that is second square parenthesis (“mechanical”) of the second member of Equation (52), is the contribution relating to the variation of internal energy due to the work interaction corresponding to the mechanical exergy; both chemical exergy and mechanical exergy constitute the two components of the generalized chemical exergy along any process. Indeed, as the internal energy is an additive state property, both contributions determined by mass interaction or work interaction with the external system (useful of reservoir), can occur sequentially or simultaneously to connect any pair of thermodynamic states. Hence, the first term constitutes the chemical exergy calculated alongside an isovolumic-isopotential process and the second term constitutes the mechanical exergy calculated alongside an isoentropic-isopotential process.

The meaning of the generalized chemical exergy is highlighted for an adiabatic and esoergonic reversible process for which work interaction only characterizes the thermodynamic state and no mass interaction and no heat interaction occur. This process is determined by absence of chemical entropy and thermal entropy variations (due to absence of mass interaction and heat interaction, respectively) and a non-null variation of mechanical entropy (due to occurring work interaction). As a consequence of the generalized formulation, if this adiabatic process is calculated in terms of exergy, the available energy (in the form of pressure mechanical energy withdrawable from the system) is accounted for in terms of its capability to be converted (and not directly transferred) into useful mass; the consequence is that the exergy analysis implies a lower amount if compared with the canonical method that identifies exergy exclusively with work interaction output conveyed to, and used by, the external system, as it is. In this regard, the entropic-mechanical addendum of the generalized chemical exergy, Equations (28) and (52), determines a reduction due to the work interaction undergoing the (reversible) entropy conversion that makes this work input not useful for a work-to-mass conversion into mass output.

## 7. Outcomes and Applications

The domain of applications of the generalized chemical exergy spreads to inorganic and organic chemistry including metabolic biological processes in living organisms. Metabolic processes determine morphological development and homeostasis as well as energetic transduction in living cells and are subdivided into two main categories: (i) catabolic processes implying the demolition of molecule aggregates and (ii) anabolic processes aimed at building-up proteins, enzymes and other organic substances and precursors. In catabolic processes, such as glycolysis, the chemical energy of glucose is transformed into chemical energy in the form of free enthalpy of adenosine tri-phosphate (ATP) [17]. The glycolysis is subdivided in two phases: (i) storing phase and (ii) releasing phase. The ATP releases chemical energy to the d-glyceraldehide-3-phosphate and is stored in these molecules during the first phase in 5 steps; instead, during the 5 steps of the second phase of glycolysis, the same chemical energy is released back to ATP, NADH and pyruvate which are products of the whole glycolysis catabolic process. The complete glycolysis process encompasses chemical exergy storage and subsequent release and the corresponding mechanical exergy and chemical exergy characterize the bi-directional inverse and direct conversions. Three, out of ten, of these processes (1st, 3rd and 10th), are irreversible and govern the entire series of reactions. The pyruvate undergoes a subsequent aerobiotic oxidation process, followed by the Krebs cycle and ending with the oxidative phosphorylation characterized by the following final oxidation reaction: $NADH+H^{+}+1/2\;O_{2}\to NAD^{+}+H_{2}O$. The ATP is used in multiple anabolic processes and the NADH and ${\mathrm{FADH}}_{2}$, reduced electron transporters, are involved in several metabolic processes [17]. In particular, the ATP is used by living organisms to release mechanical work interaction as chemical exergy output used for locomotion, for food, recovery, defense, reproduction and all other activities needed for life. Using the generalized chemical exergy provides a method to analyze aggregates, such as amino acids, proteins, enzymes and nucleic acids, constituting molecular machines, non-cyclic or cyclic, characterized by phenomena, balances and efficiencies governed by the microscopic thermodynamics at atomic and molecular level [18,19,20,21]. In this perspective, a contribution could arise in the direction of researches focusing metabolic paths and cell membrane role [22]. This approach is in use in various diseases already undergoing studies and experimental investigations [23,24,25].

## 8. Conclusions

The main outcome of the procedure described is that the generalized chemical exergy can be expressed by the sum of the two components defined as chemical exergy and mechanical exergy:(53)$$EX^{CHEMICAL}=EX^{C}+EX^{M}=W_{REV}^{CONVER}+W_{REV}^{TRANSF}+M_{REV}^{CONVER}+M_{REV}^{TRANSF}$$

This does not depend on a particular process adopted for its definition; thus, it can be considered as a general formulation which valid for any process, reversible or irreversible, connecting two different thermodynamic states.

Another outcome is that the generalized chemical exergy is determined by the equality of pressure, in addition to the equality of chemical potential, as a further condition of mutual stable equilibrium between system and reservoir. In the perspective of implications of this additional condition and the generalization to any system (large and small) in any state (equilibrium and non-equilibrium), the concept of generalized chemical exergy would require the reference to a mechanical reservoir behaving at constant pressure in addition to the chemical reservoir.

In the framework of the Gyftopoulos and Beretta perspective, the formulation of chemical entropy can be expressed in the following form adopting the symbol E to denote energy and Ω to denote available energy [2]:(54)$${\left(S_{1}-S_{0}\right)}^{C}=\frac{1}{\mu_{R}}{\left[\left(E_{1}-E_{0}\right)-\left(\Omega_{1}^{R}-\Omega_{0}^{R}\right)\right]}^{C}$$
where, if the concept of mechanical reservoir is introduced, and the equality of pressure between the system and the mechanical reservoir is considered as an additional condition for mutual stable equilibrium between system and reservoir, it remains valid. The additive property of entropy would lead to assuming that [2,13]:(55)$${\left(S_{1}-S_{0}\right)}^{M}=\frac{\bar{R}}{P_{R}V_{R}}{\left[\left(E_{1}-E_{0}\right)-\left(\Omega_{1}^{R}-\Omega_{0}^{R}\right)\right]}^{M}$$
where the mechanical component of entropy would be defined with reference to a mechanical reservoir at constant pressure.

Finally, the additivity of entropy components allows stating the following:(56)$${\left(S_{1}-S_{0}\right)}^{G}={\left(S_{1}-S_{0}\right)}^{C}+{\left(S_{1}-S_{0}\right)}^{M}$$

This should be proved to complete the formulation of generalized entropy which takes into account the general definitions proposed for chemical entropy and mechanical entropy.

As a conclusion of the present research, the methodology adopted has achieved a result for the chemical aspect that can be considered homologous to the result in the procedure already adopted for the thermal aspect and mentioned at the outset of this treatise. The equality of chemical potentials, as a condition of mutual stable equilibrium in addition to the equality of temperature and pressure of the composite system-reservoir, is an important result. Hence, the set of all conditions of mutual stable equilibrium enables establishing a more complete formulation of generalized exergy with the contribution of chemical, thermal and mechanical exergy related to a ‘thermo-chemical-mechanical’ reservoir. As a consequence, the definition of chemical entropy has been derived in relation with the molecular geometry of any system in any state, including non-equilibrium. In consideration of the importance of thermodynamic methods in chemistry and biology [26,27,28,29], different studies and applications have been developed focusing extrema principles and constructal laws [30,31,32]. In this regard, it would be worth thinking and fostering a line of research aimed at building up a rational and systematic paradigm including thermodynamic and informational aspects both constituting of intrinsic fundamentals of systems and phenomena associated with life.

## Figures and Tables

**Figure 1 entropy-23-00972-f001:**
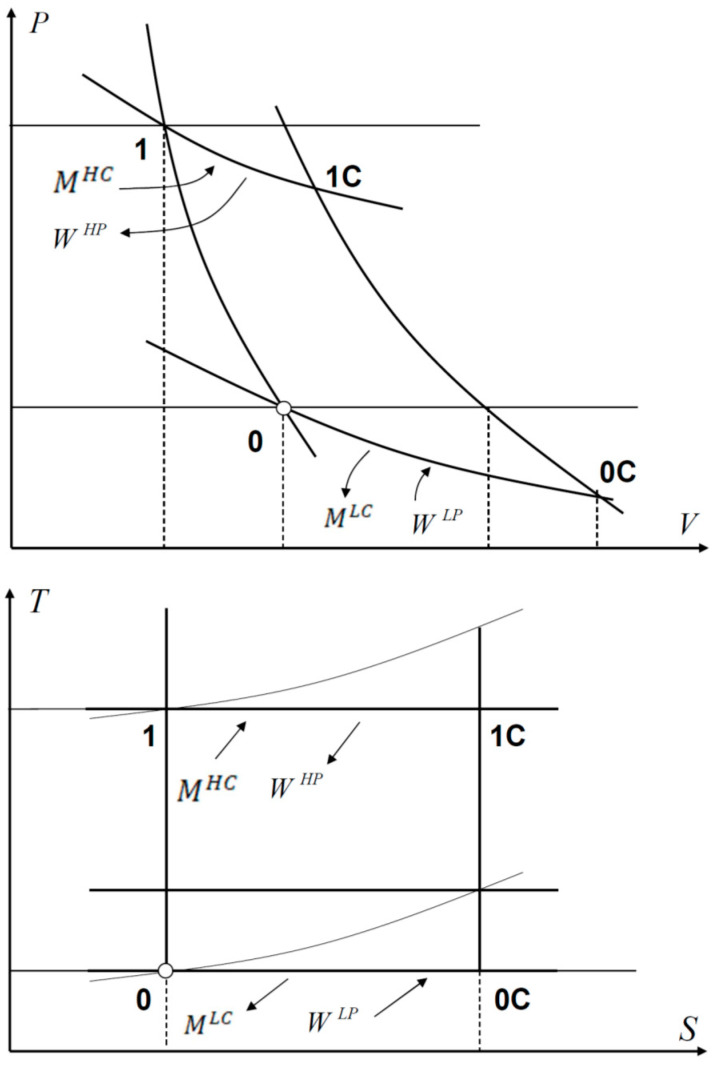
Carnot Chemical Cycle.

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
