# Peer review of "Chemical and Mechanical Aspect of Entropy-Exergy Relationship"

_entropy, 2021, doi:10.3390/e23080972_

Round 1

Author Response

Please see as attached

Reviewer 2 Report

The paper develops an analysis of the chemical aspect of exergy. The paper is interesting, but it can be improved as follows:

  • The mass interaction must be better explained
  • The Review section must be improved: I suggest to consider some more papers of Goran Wall, Sciubba, Grisolia, Demirel, Bejan, Valero, Beretta, etc.
  • In some equations, the index M isn't appropriate due to the use of M as a physical quantity in the equations themselves: i suggest to explain better the symbols used in this case
  • In Section 7, some numerical results could be useful
  • I suggest also to explain better the meaning of the arrow in the equations: this is the same symbols used by Beretta, but it could be useful to highlight better the meaning

Author Response

Please see as attached

Round 2

Reviewer 1 Report

I think the author revised the manuscript satisfactorily.

Reviewer 2 Report

I suggest to accept the paper in the present form.